# Probabilistic Seismic Hazard Analysis for Fault Dislocation Magnitude Induced by Strong Earthquakes: A Case Study of the Sichuan-Yunnan Region

**Xingwei Fu [1], Yalina Ma [2,3] and Zhen Cui [2,3,*]**

1. China Three Gorges Corporation, Beijing 100038, China; fu_xingwei@ctg.com.cn
2. State Key Laboratory of Geomechanics and Geotechnical Engineering, Wuhan 430071, China; mayalina@126.com
3. Institute of Rock and Soil Mechanics, Chinese Academy of Sciences, Wuhan 430071, China
* Correspondence: zcui@whrsm.ac.cn

**Abstract:** Seismic risk in this region is high in Sichuan-Yunnan region of western China, and active faults are well developed in this region. Tunneling in this region would inevitably come across with active faults, and the stability of the tunnel would face serious threats due to the dislocation of an active fault. The magnitude of the fault dislocation is an important design parameter in the design work of tunnels across an active fault. However, traditionally this parameter is estimated as a deterministic value that is often overestimated. In this paper, the probabilistic analysis method was introduced into the estimation of the dislocation magnitude for a tunnel in Sichuan-Yunnan region. To demonstrate the proposed approach, the Xianglushan tunnel in the Central-Yunnan-Water-Transmission Project, which crosses the Longpan–Qiaohou active fault, was taken as an example case. The seismicity parameters in Sichuan-Yunnan region and the dislocation probability caused by earthquakes are considered. The fault dislocation magnitude that may occur during the service life of the tunnel can be estimated as a probability function, and parameter analysis was conducted. Thus, anti-dislocation design work of the tunnel may be conducted based on this reasonable estimation.

**Keywords:** tunnel; active fault zone; dislocation magnitude; seismicity parameter; probabilistic analysis method

## 1. Introduction

Traditionally, design and construction codes of tunnels recommend avoiding tunneling in the vicinity of active faults and further suggest a certain avoidance distance [1,2]. However, at times, due to the limitation of route selection, especially in the high seismic risk regions such as southwest China, tunnels crossing active faults are inevitable to be faced with the potential damage of fault dislocation.

Previous research has shown that tunnels are vulnerable to seismic fault zones with poor quality of the surrounding rock mass and significant changes in the stratigraphic type. Typical cases of seismic damage of tunnels across faults are shown in Table 1. These cases highlight the importance of the study of tunnel anti-dislocation design, and the magnitude of fault dislocation is the important input parameter for the anti-dislocation design. However, at present, limited studies exist considering fault dislocation magnitude in the anti-dislocation design of cross-fault tunnels, and none of them have considered the tunnel's seismic performance demands. The determination of fault dislocation magnitude in the anti-fault design should be based on economic and safety. Unnecessary reinforcement and endurable construction cost would happen if the magnitude was overestimated. On the other hand, failure risk would emerge once the dislocation magnitude was underestimated.

At present, the hazard analysis for fault dislocation has generally performed with laboratory experiments [3–5] and numerical analysis [6,7]. Cole et al. [3] carried out sandbox

tests in 1984 and summarized the influence of overburden rock thickness, fault dip, and other factors on fault dislocation. Scott et al. [6] studied the response behavior of bedrock under vertical displacement with a numerical model. In addition, hazard analysis for fault dislocation can also be conducted with statistical methods [8–10], i.e., based on seismic damage data and statistical regression methods, the relationship between earthquake magnitude and surface rupture length, surface displacement could be established. With this approach, Toeher [8] proposed a statistical relationship between earthquake magnitude and fault rupture. Wells et al. [9] conducted statistical analysis based on historical records worldwide and established the relationship between surface displacement and fault dislocation under various earthquake magnitudes. Generally, the abovementioned methods are based on the statistical data of the fault rupture length under given seismic conditions. They are essentially deterministic methods that analyze the dislocation probability in a certain period of time and evaluate the maximum dislocation magnitude. With these methods, the anti-dislocation measure of a tunnel was to be design according to the target fault's maximum potential dislocation magnitude. Thus, the randomness of earthquake occurrence and that of the fault rupture can be hardly considered. Besides, this deterministic method for fault dislocation magnitude induced by strong earthquakes is difficult to coordinate with the widely used probabilistic seismic hazard analysis (PSHA). Hence, in order to consider the randomness of earthquake occurrence and the uncertainty of seismicity parameters, it is desirable to extend the PSHA into the fault dislocation estimation, which is important for the anti-dislocation design of cross-active fault tunnels.

**Table 1.** Typical cases of cross-fault tunnel seismic damages. [11–15].

| No. | Year | Event | Moment Magnitude/$M_w$ | Description of the Tunnel Damage |
|---|---|---|---|---|
| 1 | 1906 | San Francisco earthquake, USA | 8.3 | The San Andreas dam catchment tunnel crossing the fault zone was seriously deformed by 2.4 m. The Wright No. 1 tunnel had a phenomenon of orbital uplift, the horizontal displacement was 1.37 m. |
| 2 | 1930 | Izu earthquake, Japan | 7.3 | The sidewall of the Danah railway tunnel cracked seriously. The horizontal displacement reached 2.39 m and the vertical displacement reached 0.6 m. |
| 3 | 1971 | San Fernand earthquake, USA | 6.4 | The liner of the San Fernando tunnel near the Sylmar fault was damaged and displaced on a large scale. |
| 4 | 1995 | Ojima earthquake, Japan | 7.2 | The inverted arch and sidewall of the Inatori tunnel were cracked under the fault rupture, and the concrete of the vault roof was peeled off. |
| 5 | 1999 | Chi-chi earthquake, Taiwan, China | 7.3 | Under the dislocation of the Chelongpu fault, the Shigangba tunnel was damaged seriously near the fault plane. The vertical deformation of the tunnel reached 4.0 m and the horizontal deformation was up to 3.0 m. |

In this paper, considering the seismicity parameters and the fault dislocation probability induced by strong earthquakes, the PSHA for fault dislocation magnitude induced by strong earthquakes was introduced. In this way, the fault dislocation is associated with seismicity, which can reflect the randomness nature of earthquake event occurrence. With this approach, the fault dislocation distance during the tunnel's service life can be expressed as a function of exceedance probability. As a contrast, traditionally, this dislocation distance was estimated as a deterministic value. The term "the probability of dislocation distance" was used to express the estimation results. Ultimately, the parameters analysis was carried out for the example project, the Xianglushan tunnel of the Central-Yunnan-Water-Transmission Project that crosses the Longpan–Qiaohou active fault. The conclusions obtained may provide some references for the anti-dislocation issue of tunnels across active faults.

## 2. Principle of the PSHA for Fault Dislocation Estimation

### 2.1. The Dislocation Probability Induced by Strong Earthquakes

The PSHA of potential fault dislocation is to calculate the probability that the dislocation distance $U$ may reach or exceed the engineering allowable value $u$ of a site along the fault track within the service life $T$ of the project.

Ma et al. [16] made statistics on the surface displacement of 340 strong earthquake events with magnitude >6 in mainland China since 1900, and confirmed that the surface displacement was generally caused by strong earthquakes. However, a strong earthquake event is not necessarily accompanied with surface displacement. Thus, the frequency of surface displacement is much lower than that of the strong earthquake occurrence. The dislocation probability induced by a strong earthquake event is expressed by $P_M$, obtained by Equation (1):

$$P_M = \int_{m_0}^{m_1} f(m) \cdot f(M) \mathrm{d}m \tag{1}$$

where $m_0$ and $m_1$ are the floor and ceiling of earthquake magnitude to be evaluated. $f(M)$ is the probability density function (PDF) of the earthquake magnitude $M$, which is determined by seismicity parameters, in the following forms:

$$f(M) = \frac{\beta \cdot \exp(-\beta \cdot (m - m_0))}{1 - \exp(-\beta \cdot (m_1 - m_0))}, \ m_0 \leq m \leq m_1 \tag{2}$$

where $\beta$ is the distribution function of the magnitude, which refers to the coefficient of distribution followed by earthquakes with magnitude $m_0 \leq m \leq m_1$ occurred in a potential seismic zone. $\beta$ can be derived from the Gutenberg-Richter model $\lg N = a - bM$ [17], expressed as $\beta = b \cdot ln10$.

$f(m)$ is a conditional PDF of the fault dislocation induced by the earthquake of magnitude $M$, which is related to magnitude, focal depth, and thickness of overburden, and with great uncertainty. Generally, according to the relationship between the frequency of surface displacement and earthquake magnitude $M$ in historical data, a statistical regression equation can be obtained with the magnitude of $M$ as a variable.

### 2.2. The Probability That the Dislocation Magnitude Exceeding a Given Value

Based on the dislocation probability induced by earthquake and considering regional seismicity parameters, the probability $P(U \geq u)$ that the dislocation magnitude exceeding a given value on a site of fault was proposed by [18], expressed as Equation (3):

$$P(U \geq u) = P(\text{a site affected by fault disloaction} | E, e)$$
$$\times \int_0^\infty \int_u^{u_1} \int_m^{m_1} P(\text{a site affected by dislocation distance of } U | E_U, e) \times f(u) \times P_M \times f(M) \mathrm{d}e \mathrm{d}u \mathrm{d}m \tag{3}$$

where $P(\text{a site affected by fault disloaction} | E, e)$ is the probability of an engineering site affected by the dislocation of a certain fault under a seismic event $E$, which can be calculated according to the ratio of the target fault length to the total length of active faults in the seismic zone.

$P(\text{a site affected by dislocation distance of } U | E_U, e)$ is the probability that dislocation would occur for a specific engineering site, under a certain earthquake magnitude of $E_U$ that produced the dislocation distance of $U$, and random error of $e$. It is related to the location of the engineering site on the fault, and also related to the boundary conditions of the fault.

Based on historical statistics, the boundary condition is assumed that an earthquake may occur at any location on a fault, but that the fault dislocation cannot exceed the fault endpoint. Then the probability that the dislocation $u$ affecting the site is described in Equation (4).

$$P(\text{a site affected by dislocation distance of } U | E_U, e) = (\varphi \cdot u \cdot e) / L \tag{4}$$

where $\varphi = \exp(c - a \cdot d / b)$. $a$, $b$, $c$ and $d$ are regression coefficients, which can be estimated from empirical formulas $\ln U = a + bM$, $M = a_1 + b_1 \cdot \lg(U_{max})$, $\ln S = c + dM$, and $M = c_1 + d_1 \cdot \lg(S_{max})$, respectively. $U_{max}$ is the maximum possible dislocation magnitude and the $S_{max}$ is maximum possible dislocation length along the fault.

$f(u)$ is the probability density function of the fault dislocation distance $u$ under the earthquake of magnitude $M$, expressed as Equation (5):

$$f(u) = k \cdot u^{-v}, U_0 \leq u \leq U_1 \tag{5}$$

where $k = \beta / b \cdot \left( u_0^{-\beta/b} - u_1^{-\beta/b} \right)$, $v = \beta / b + 1$, $u_0 = \exp(a + bm_0)$, $u_0 = \exp(a + bm_1)$, and $\beta$ is the magnitude distribution function mentioned above.

### 2.3. The Probability That the Dislocation Distance Exceeding a Given Value in T-Year

The probabilistic analysis method of fault dislocation is assumed to be a uniform Poisson's process with average annual incidence $v$. Then, the probability $P_T$ that the dislocation distance $U$ exceeding a given value $u$ of a site in $T$-year can be expressed as Equation (6) [18]:

$$P_T = 1 - \left( \exp(-P(U \geq u)) \cdot v \right)^T \tag{6}$$

where $v$ is the average annual incidence of earthquakes. It is the number of earthquakes with magnitude greater than or equal to $m_0$ occurring annually on a fault. $P_T$ is obtained by bringing the $P(U \geq u)$ mentioned in Section 2.2 into Equation (6).

### 3. Fault Dislocation Estimation in Sichuan-Yunnan Region

#### 3.1. Seismicity in Sichuan-Yunnan Region

The Sichuan-Yunnan region roughly refers to the area composed of Sichuan Province and Yunnan Province (illustrated in Figure 1). The region is located to the east of the Tibet Plateau and is known for intense tectonic movement and numerous active faults. It is the most high-risk seismic region in China. In addition, there are high frequency and intensity of seismicity in this region. Thirty-two earthquakes with magnitude 7 or above have been recorded, of which two have exceeded magnitude 8. These earthquakes are mainly strike-slip earthquakes with focal depths of 10–15 km, and they have caused heavy property losses and casualties [19–21].

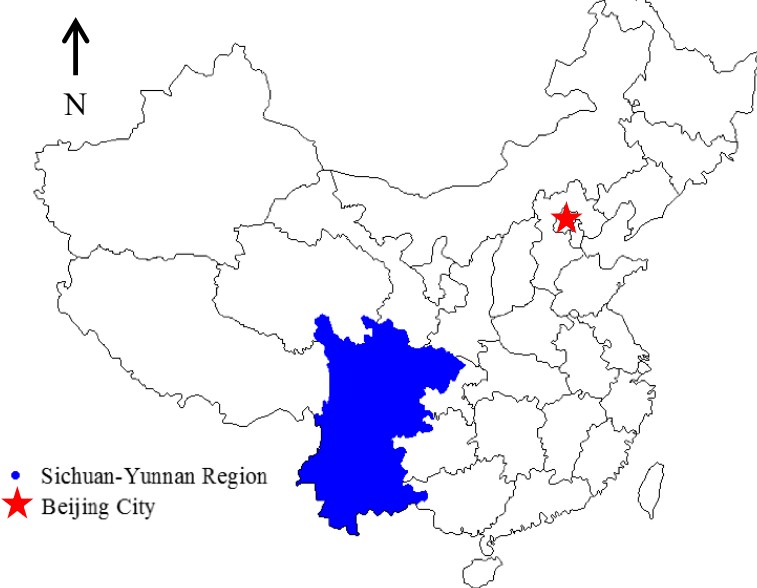

**Figure 1.** Location of Sichuan-Yunnan region.

It is a known fact that earthquakes with the magnitude $\geq 6.5$ are often accompanied by the formation of surface rupture zones [22–24]. Deng et al. [25] confirmed that the surface rupture was generally accompanied by the earthquake of magnitude $\geq 6.5$ in mainland China. Therefore, the destructive earthquakes studied in this paper are limited to earthquakes of magnitude 6.5 to 8. Typical examples of surface displacement induced by strong earthquakes in the Sichuan-Yunnan region are shown in Table 2.

**Table 2.** Typical examples of surface displacement induced by strong earthquakes in Sichuan-Yunnan region [16,26].

| No. | Date | Location | Moment Magnitude/$M_w$ | Earthquake Damage |
|---|---|---|---|---|
| 1 | 25-May-1948 | Litang, Sichuan | 7.3 | The ground fissures were in goose-shaped arrangement, extending northwest and exposing intermittently. |
| 2 | 14-Apr-1955 | Kangding, Sichuan | 7.5 | The depth of mountain fissures was 2 m, the length of flat ground fissures was 40–50 m, and the width was 1 m. |
| 3 | 9-Nov-1960 | Songpan, Sichuan | 6.75 | Ground fissures occurred intermittently, especially on hillsides and ridges. They were mostly tens of meters in length and 3–5 cm in width. |
| 4 | 5-Feb-1966 | Dongchuan, Yunnan | 6.5 | Ground fissures were mostly banded and intermittently exposed, with a length of 10–50 m and a width of 3–8 cm. |
| 5 | 5-Jan-1970 | Tonghai, Yunnan | 7.7 | The cracks on the hillside were distributed in goose shape, with good continuity and a few meters to tens of meters wide. Dextral strike-slip fault, the horizontal displacement was nearly 1 m. |
| 6 | 6-Feb-1973 | Luhuo, Sichuan | 7.9 | The fault dislocation occurred on the bedrock slope with a width of 1.5 m, horizontal displacement of 3.6 m and vertical displacement of 2 m. |
| 7 | 15-Mar-1979 | Puer, Yunnan | 6.8 | The largest single crack was up to 100 m long and 20 cm wide, with a direction of N50° W. |
| 8 | 6-Nov-1988 | Gengma, Yunnan | 7.2 | The fault dislocation occurred on the hillside and was intermittently exposed. The maximum horizontal twist was 72 cm, the vertical displacement was 60 cm, and the crack width was about 30 cm. |
| 9 | 6-Sep-1993 | Yanglin, Yunnan | 8 | The Xiaojiang fault was intermittently exposed with a fracture width of about 1.5 m and a vertical displacement of about 1 m. |

### 3.2. Seismic Parameters of the Example Site

The Longpan–Qiaohou active fault crossed by the Xianglushan tunnel of the Central-Yunnan-Water-Transmission Project, is taken as the example case here. Xianglushan tunnel is located in the Xianshuihe–Diandong seismic zone of the Qinghai-Tibetan region, which contains many problematic areas, including some complicated formations, areas of tectonics and severely faulted zones. There are multiple faults intersect with the tunnel axis, and three of them are the active faults, namely, Longpan–Qiaohou fault, Lijiang–Jianchuan fault, and Heqing–Eryuan fault, as shown in Figure 2a. Among them, the Longpan–Qiaohou fault is the widest and is a strike-slip fault (see Figure 2b). The orientation of the fault is approximately N10° E NW∠80°, with the length of 240 km, which pose a serious threat to the tunnel. The fault started at least since the early Paleozoic, showing the complex nature of long-term and multi-phase activities. There have been many earthquakes of magnitude >5 recorded, with two major events of magnitude $6\frac{1}{4}$. It is one of the most important seismogenic structures in northwestern Yunnan [27–29].

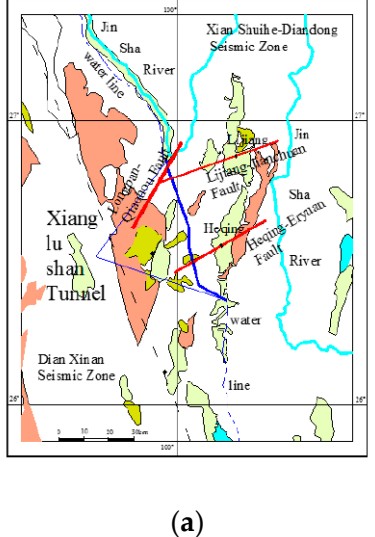

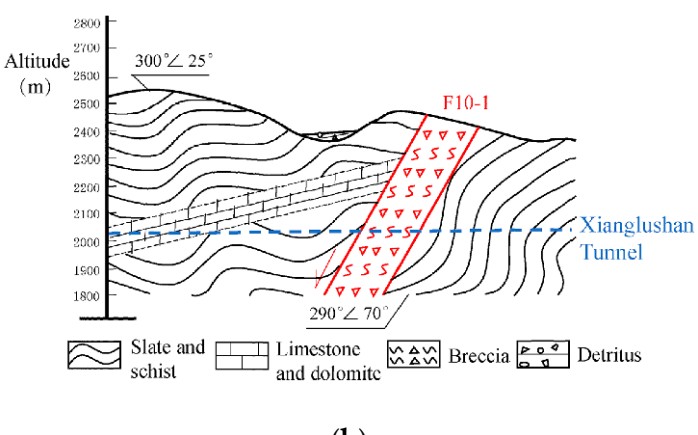

(**a**)             (**b**)

**Figure 2.** (**a**) Location of Xianglushan tunnel. (**b**) Geological longitudinal profile of the Xianglushan tunnel and the Longpan–Qiaohou fault.

To conduct the PSHA-based fault dislocation estimation for this region, the seismicity parameters must be provided. Floor magnitude $m_0$ was determined to be 6.5 since no notable surface rupture was reported in the local historical events (maximum magnitude = $6\frac{1}{4}$). The ceiling magnitude $m_1$ was determined to be 8.0 following the existing literatures [30,31]. The existing literatures also suggested a $\beta = 1.74$ and $v_{6.5} = 1$ for the Sichuan-Yunnan region. The seismicity parameters are listed in Table 3.

**Table 3.** Seismicity parameters of the area.

| Parameters | $m_0$ | $m_1$ | $v_{6.5}$ | $\beta$ |
|---|---|---|---|---|
| Value | 6.5 | 8.0 | 1 | 1.74 |

### 3.3. The Dislocation Probability Induced by Strong Earthquakes

$P_M$ is the dislocation probability induced by strong earthquakes, as described in Section 2.2. When the earthquake magnitude $M$ belongs to the range of $m_0 \leq m \leq m_1$,

$$f(M) = \frac{\beta \cdot \exp(-\beta \cdot (m - m_0))}{1 - \exp(-\beta \cdot (m_1 - m_0))} = \frac{1.74 \cdot \exp(-1.74 \cdot (m - 6.5))}{1 - \exp(-1.74 \cdot (8 - 6.5))} \approx \frac{153394}{\exp(-1.74m)} \quad (7)$$

$f(m)$ is a conditional probability density function of fault dislocation induced by earthquake of magnitude $M$. Ma et al. [16] suggested a probability function of fault dislocation under various seismic magnitude for the western region of China, as illustrated as the solid line in Figure 3. Here, however, to simplify the formula derivation work in the following text, a simplified equation with earthquake magnitude as variable under the condition of magnitude 6.5 to 8 was obtained by the statistical regression method as $f(m) = 47.5m - 300$, while $6.5 \leq m \leq 8$, as the dashed line in Figure 3.

Substituting the above calculation results into $P_M = \int_{m_0}^{m_1} f(m) \cdot f(M) \mathrm{d}m$, the following can be obtained:

$$P_M = \int_{6.5}^{8} f(47.5m - 300) \cdot \frac{153394}{\exp(-1.74m)} \mathrm{d}m \approx 0.3 \quad (8)$$

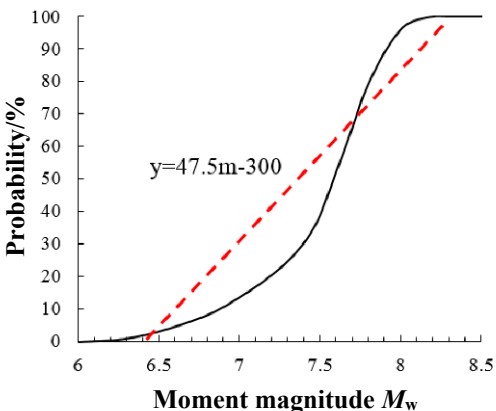

**Figure 3.** The probability of fault dislocation under various seismic magnitude.

*3.4. The Probability That the Dislocation Magnitude Exceeding a Given Value*

As mentioned above:

$$P(U \geq u) = P(\text{a site affected by fault disloaction}|E, e)$$
$$\times \int_0^\infty \int_u^{u_1} \int_m^{m_1} P(\text{a site affected by dislocation distance of } U|E_U, e) \times f(u) \times P_M \times f(M) \mathrm{d}e \mathrm{d}u \mathrm{d}m \tag{9}$$

Step 1: The probability of an engineering site affected by fault dislocation can be calculated according to the ratio of the target fault length to the total length of active faults in the study zone.

$$P(\text{a site affected by fault disloaction}|E, e) = \frac{\text{length of the target active fault}}{\text{total length of active faults in the study area}} = \frac{240}{408} = 0.408 \tag{10}$$

Step 2: By calculating with the empirical formula: $\ln U = a + bM$, $M = 7.205 + 0.974 \cdot \lg(U_{max})$ [25], the result of the calculation is $a = -17.044$, $b = 2.365$. Substituting $a$ and $b$ into the formula, we can get:

$$u_0 = \mathrm{e}^{a+bm_0} = \mathrm{e}^{-17.044+2.365 \cdot 6.5} \approx 0.187, \text{ and } u_1 = \mathrm{e}^{-17.044+2.365 \cdot 8} \approx 6.5 \tag{11}$$

Next, by calculating with the empirical formula: $\ln S = c + dM$, and $M = 5.11 + 0.579 \lg(S_{max})$ [32,33], it can be calculated that $c = -20$, $d = 3.9$, then,

$$\varphi = \exp(-20 - (-17.044 \cdot 3.9/2.365)) \approx 5901 \tag{12}$$

Thus,

$$P(\text{a site affected by dislocation distance of } U|E_U, e) = \frac{5901 \cdot u \cdot e}{240 \text{km}} \approx 0.025 \cdot u \cdot e \tag{13}$$

Step 3: $f(u) = k \cdot u^{-v}$, then

$$k = 0.76 \cdot \left( 0.187^{-0.76} - 6.5^{-0.76} \right)^{-1} \approx 0.23 \tag{14}$$

$$v = 1.74/2.365 \approx 1.76 \tag{15}$$

Thus,

$$f(u) = k \cdot u^{-v} = 0.23 \cdot u^{-0.76} \tag{16}$$

Step 4: $P_M$ is calculated to be 0.3 from the previous calculation in Section 3.4.

Step 5: When $6.5 \leq m \leq 8$, we can get:

$$f(M) = \frac{\beta \cdot \exp(-\beta \cdot (m - m_0))}{1 - \exp(-\beta \cdot (m_1 - m_0))} = \frac{153394}{\exp(-1.74m)} \tag{17}$$

Thus,

$$P(U \geq u) = 0.408 \times \int_0^\infty \int_u^{6.5} \int_{6.5}^8 0.25 \cdot u \cdot e \cdot 0.23 \cdot u^{-1.76} \cdot 0.3 \cdot 153394 / \exp(-1.74 \cdot m) \mathrm{d}e \mathrm{d}u \mathrm{d}m \qquad (18)$$

The random error $e$ is ignored here for concision and simplicity.

### 3.5. The Probability That the Dislocation Magnitude Exceeding a Given Value in T-Year

The probability $P_T$ that the dislocation magnitude $U$ exceeding a given value $u$ of a site in $T$-year can be expressed as Equation (6) mentioned in Section 2.3:

$$P_T = 1 - (\exp(-P(U \geq u)) \cdot v_{6.5})^T \qquad (19)$$

Thus, the dislocation probability of the Longpan–Qiaohou fault within the tunnel's 50-year service life is:

$$P_{50} = 1 - (\exp(-P(U \geq u)) \cdot 1)^{50} = 1 - \frac{1}{\exp(P(U \geq u) \cdot 1 \cdot 50)} \qquad (20)$$

The dislocation probability of the Longpan–Qiaohou fault within 100 years is:

$$P_{100} = 1 - (\exp(-P(U \geq u)) \cdot 1)^{100} = 1 - \frac{1}{\exp(P(U \geq u) \cdot 1 \cdot 100)} \qquad (21)$$

### 3.6. The Probability That the Dislocation Magnitude Exceeding a Given Value in T-Year

According to the abovementioned PSHA method for fault dislocation magnitude induced by strong earthquakes, the calculation results are listed in Table 4 and Figure 4. As can be seen from the figure, the probability level of fault dislocation decreases with the increase of dislocation magnitude. The probability of exceedance of 63% within 50 years corresponds to the Frequent Earthquake Level, and the dislocation magnitude is about 0.1m. Meanwhile, the probability of exceedance of 10% within 50 years corresponds to the Moderate Earthquake Level, and the dislocation magnitude is about 1.2 m. The probability of exceedance of 2%~3% within 50 years corresponds to the Rare Earthquake Level, and the dislocation magnitude is about 2.7~3.3 m. According to the design code of the tunnel, the anti-dislocation design of the tunnels was conducted based on the Moderate Earthquake Level, and the corresponding dislocation magnitude is about 1.2 m [34].

**Table 4.** Dislocation distance of the Longpan–Qiaohou fault with different probability.

| Probability of Exceedance within 50 Years | 63% | 10% | 3% | 2% |
|---|---|---|---|---|
| Dislocation magnitude/m | 0.1 | 1.2 | 2.7 | 3.3 |

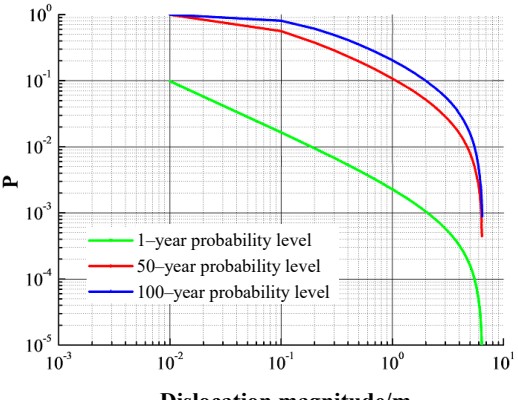

**Figure 4.** Relationship between the exceedance probability level *P* and dislocation magnitude of the Longpan–Qiaohou fault.

## 4. Parameter Study

### 4.1. Average Annual Incidence of Earthquakes $v$

The average annual incidence of earthquakes $v$ refers to the average annual number of earthquake events of magnitude $m_0 \leq m \leq m_1$ in a seismic zone, which was related to the probability density function $f(M)$. In addition, the occurrence of earthquakes follows the uniform distribution in seismic zones, and $v$ is a constant. Combined with historical data of Ma et al. [16] and Wang et al. [35], the average annual incidence of earthquakes in the study regions are listed in Table 5. Here, the parameter analysis was carried out for the average annual incidence of earthquakes $v$. The results are shown in Figure 5.

**Table 5.** Average annual incidence of earthquakes $v$.

| Study Region | Magnitude | Average Annual Incidence of Earthquakes $v$ |
| --- | --- | --- |
| North Tianshan area [35] | $4 \leq m \leq 8$ | $v_{4.0} = 9$ |
| Mainland China [10] | $6 \leq m \leq 8$ | $v_{6.0} = 4$ |
| Sichuan-Yunnan region (Current work) | $6.5 \leq m \leq 8$ | $v_{6.5} = 1$ |

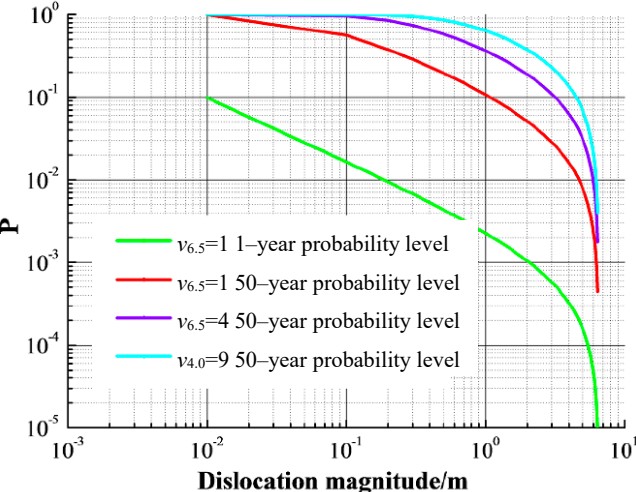

**Figure 5.** Relationship between the probability level $P$ and dislocation magnitude under different $v$.

Since the average annual incidence of earthquakes only affects the relationship between the probability level and the dislocation magnitude for multiple years, the 50-year probability level was taken as the study object. As can be seen from Figure 5, under the different value of $v$, the trend of the curves of probability level vs. dislocation magnitude is similar. When the dislocation magnitude is in the range of 0.01 m to 6.5 m, the 50-year probability level decreases with the decreasing $v$.

### 4.2. Magnitude Distribution Function $\beta$

$\beta$ is the magnitude distribution function, which was derived from Gutenberg-Richter model and was introduced in Section 2.1. In the relationship of magnitude-frequency, $\beta$ is the negative value of the gradient [35]. Meanwhile, $\beta$ has a certain influence on the probability density function of the magnitude ($f(M)$), the dislocation probability induced by strong earthquakes ($P_M$), and the probability density function of the fault dislocation magnitude $u$ under the earthquake of magnitude ($f(u)$). In this section, based on the magnitude-frequency relationship of mainland China [36,37], Sichuan-Yunnan region, and North China, $\beta$ of different research areas were obtained (Table 6). Parameters study for $\beta$ was carried out. Results are shown in Figure 6.

**Table 6.** Magnitude density distribution coefficient $\beta$ [31,32,38].

| Study Area | Magnitude | $\beta$ |
|---|---|---|
| Mainland China | $\lg N = 5.14 - 0.82M$ | 1.90 |
| Sichuan-Yunnan region | $\lg N = 4.21 - 0.76M$ | 1.74 |
| North China | $\lg N = 5.27 - 0.65M$ | 1.50 |

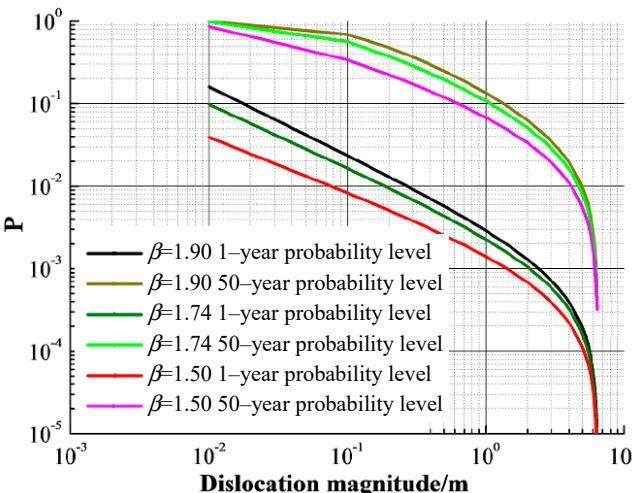

**Figure 6.** Relationship between the probability level *P* and dislocation magnitude under different $\beta$.

The 1-year and 50-year probability levels are taken as study objects. As can be seen from Figure 6, the trend of the curves of probability level vs. dislocation magnitude is similar under different $\beta$. Under the same fault dislocation magnitude, the 1-year probability level decreases with the decreasing $\beta$. When the dislocation magnitude is less than decimeter magnitude, the 1-year probability level differs greatly with different $\beta$. In addition, when the dislocation magnitude is in the range of centimeters to meters, the 50-year probability level decreases with the decrease of $\beta$. The above findings indicates that the results estimated with the $\beta$ value of Sichuan-Yunnan region are more suitable for the Xianglushan tunnel project.

*4.3. Conditional Probability Density Function $f(m)$*

$f(m)$ is the conditional probability density function of the fault dislocation induced by earthquake of magnitude *M*, obtained by statistical regression based on historical data. It has an effect on dislocation probability induced by strong earthquakes $P_M$. As mentioned in Section 3.3, the equation of magnitude-probability with earthquake magnitude as variable under the condition of magnitude 6.5 to 8 in western China was obtained. In this section, according to the existing statistical data for mainland China, western China, and northern China, the relationship between the probability of surface rupture and the earthquake magnitude summarized. Thus, the conditional probability density function $f(m)$ of mainland China, western China, and northern China were obtained, as list in Table 7. Results of parameters study are illustrated in Figure 7.

**Table 7.** Conditional probability density function induced by magnitude *M*.

| Study Area | Conditional Probability Density Function *f(m)* | Probability of Earthquake-Induced Dislocation $P_M$ |
|---|---|---|
| Mainland China | $y = 50m - 300$ | 0.47 |
| Western China | $y = 47.5m - 300$ | 0.30 |
| North China | $y = 66m - 450$ | 0.10 |

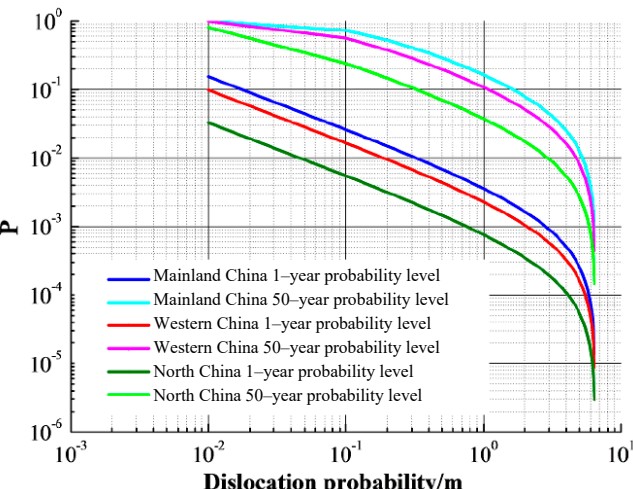

**Figure 7.** Relationship between the probability level $P$ and dislocation magnitude under different $f(M)$.

It can be seen from Figure 7 that the trend of the curves of probability level vs. dislocation magnitude is similar under different $f(M)$ in the different regions. Under the same fault dislocation magnitude, the probability level of mainland China is similar to that of Western China, but greater than that of North China. Again, the above findings indicates that the results estimated with the $f(M)$ of Sichuan-Yunnan region are more suitable for the Xianglushan tunnel project.

## 5. Conclusions

In this paper, the PSHA for fault dislocation magnitude induced by strong earthquakes was applied to a tunnel in Sichuan-Yunnan region. The regional seismicity parameters of the Sichuan-Yunnan region and the dislocation probability caused by local earthquakes were considered. On the basis of these analyses, the following conclusions are drawn:

(1) In the anti-dislocation design of tunnels crossing active faults, the magnitude of fault dislocation is the important input parameter. For example, when the dislocation magnitude is small, no special design is required other than some strengthening in the tunnel liner. However, if the estimated dislocation magnitude is significant, special anti-dislocation design is required, e.g., increasing the tunnel span. Generally, fault dislocation was caused by strong earthquakes. However, strong earthquakes are not necessarily accompanied with fault dislocation. The determination of the dislocation probability caused by strong earthquakes is a prerequisite for probabilistic seismic hazard analysis. Thus, the PSHA-based dislocation estimation should consider floor magnitude of the specific region.

(2) In the PSHA method for fault dislocation magnitude induced by strong earthquakes adopted in current study, multiple factors were considered. In this manner, the fault dislocation is associated with regional seismicity of the Sichuan-Yunnan region, which would bring more reasonable results for the example engineering case.

(3) According to the PSHA for Longpan–Qiaohou active fault crossed by the Xianglushan tunnel, it can be obtained that the exceedance probability of fault dislocation decreases with the increasing dislocation magnitude. The estimated dislocation magnitude is about 0.1 m under the exceedance probability level of 63% within 50 years (Frequent Earthquake Level). The estimated dislocation magnitude is about 1.2m under the exceedance probability level of 10% within 50 years (Moderate Earthquake Level). The dislocation is 2.7~3.3 m for the Rare Earthquake Level (exceedance probability level of 2%~3% within 50 years). According to the design code of the tunnel, the anti-dislocation design of the tunnels was conducted based on the dislocation magnitude of Moderate Earthquake Level, which is 1.2 m.

(4) Under the same fault dislocation magnitude, the probability level was influenced by the average annual incidence of earthquakes $v$, the magnitude distribution function $\beta$, and the conditional probability density function of the fault dislocation induced by earthquake of magnitude $M$, namely, $f(m)$. In the current study, with parameter study, characteristics of the fault dislocation in Sichuan-Yunnan region were compared with that of the entire China. Significant differences were revealed, indicating that comparing with the existing results, the results estimated with current study are more suitable for the Xianglushan tunnel project.

(5) Under the different value of the average annual incidence of earthquakes $v$, the magnitude distribution function $\beta$, and the conditional probability density function $f(m)$, the overall trend of the relationship curves between probability level and dislocation magnitude are similar with each other. The 50-year probability level would decrease with the decreasing $v$ and $\beta$, while the dislocation magnitude is in the range of centimeters to meters.

(6) The main limitations of the current study are the adoption of the simplified probability function of fault dislocation under various seismic magnitude, and the ignoration of the random error for the dislocation probability. These drawbacks should be solved in the future work.

**Author Contributions:** The paper was written by X.F. under the guidance of Y.M. and Z.C. The formal analysis and pre-literature research were carried out by Z.C. The PSHA for fault dislocation estimation was proposed and by X.F. and Z.C. The parameter study was carried out by X.F. under the help of Y.M. All authors have read and agreed to the published version of the manuscript.

**Funding:** This study was financially supported by the National Key R&D Program of China (no. 2016YFC0401803), the National Natural Science Foundation of China, (no. 51779253, 52079133), the Key Laboratory for Geo-Mechanics and Deep Underground Engineering, China University of Mining & Technology (no. SKLGDUEK1912), CRSRI Open Research Program (program SN: CKWV2019746/KY), and the Youth Innovation Promotion Association CAS.

**Institutional Review Board Statement:** Not applicable.

**Informed Consent Statement:** Not applicable.

**Data Availability Statement:** The data presented in this study are available on request from the corresponding author.

**Conflicts of Interest:** The authors declare that they have no conflict of interest.

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
