# Peer review of "Probabilistic Seismic Hazard Analysis for Fault Dislocation Magnitude Induced by Strong Earthquakes: A Case Study of the Sichuan-Yunnan Region"

_sustainability, doi:10.3390/su13042383_

Round 1

Reviewer 1 Report

General comment:

The current draft has very silly mistakes that are not acceptable. Authors should improve the background and/or objective and where other studies have lacked in presenting this kind of analysis. Be clear in expressing the significance of the current study. Authors should also reassess sentence structure/grammar throughout the paper. Authors can proof check with the English native speaker.

The detailed comments which may be considered to improve this draft are:

  • Line 3: what is representing superscript 3? Please mention
  • Line 26-27: “The dislocation magnitude of the ….” Please rewrite it.
  • Line 40: Remove these references and provide after Table 1 heading in ascending order of the year.
  • Line 50: Arrange the references in ascending order of year here and also elsewhere in this draft such as Line 55, 160, 164.
  • Line 72: Table 1; Arrange the events in ascending order of year and which magnitude are you taking, please provide full name, not just magnitude?
  • Line 77: “The magnitude of …... rather than a constant value.” Rewrite this sentence.
  • Line 85-96: This is irrelevant content, authors can remove it.
  • Line 112 and 114 and 149: “where ??? is the distribution…. And seismic zone *** and where v xxx is the average. Provide the missing content (???, ***, xxx).
  • Line 131: It is related to the location …. conditions of the fault. Rewrite this sentence.
  • Line 152: the heading of section 3 is the same as section 2. Please check it.
  • Line 154-156: “Located in the eastern margin …. types and activities. Rewrite this sentence
  • Line 163: “are several … several meters”. Rewrite this sentence.
  • Line 168: In Figure 1, the Beijing city symbol not visible and provides the direction symbol as it is showing the map.
  • Line 172: Table 2; Arrange the events in ascending order of year and which magnitude are you taking, please provide full name, not just magnitude? Provide the references after Table 2 heading in ascending order of the year.
  • Line 134: heading is not relevant to main content; should be modified
  • Line 193: “Thus, the @@ is 1.74... missing content. Please provide.
  • Line 201: According to Ma et al. (2015) ….
  • Line 205: In figure 3, authors can use the polynomial curve fit, and the equation of y can be changed. The earthquake magnitude is which magnitude, moment, or something else please specify.
  • Line 263: remove “4” in the last.
  • Line 279: In figure 5, authors can use different line patterns (dotted, dash-dotted) for the graph to make it clearer in black and white, and what is P? Please write in the graph title.
  • Line 287: The reviewer is not getting the author's point. Please provide some background information.
  • Line 295: Remove the reference and provide after Table 6 heading in ascending order of the year.
  • Line 299: Revised Figure 6 as suggested for Figure 5.
  • Line 302: b should be in symbol
  • Line 319: remove the line “Probability level of 1-year and 50-year are taken as the study objects” as already stated at Line 300.
  • Line 356: Point 6 is not a conclusion. It can be a summary of all conclusions so remove point 6.

Author Response

First of all, the authors are grateful for the reviewers’ comments on the language problem of the current manuscript. The authors are very sorry for this. And the author would have the manuscript been language edited by MDPI in the subsequent process, after the peer review stage.

 Response to Reviewer 1 Comments

  1. Response to comment: (Line 3: what is representing superscript 3? Please mention)

We are very sorry for our carelessness. Superscript 3 stands for Institute of Rock and Soil Mechanics, CAS. And this carelessness was corrected.

  1. Response to comment: (Line 26-27: “The dislocation magnitude of the ….” Please rewrite it.)

We are sorry for our unqualified English level. These sentences have been rewritten.

  1. Response to comment: (Line 40: Remove these references and provide after Table 1 heading in ascending order of the year.)

The authors are very thankful for the viewer’s suggestion. These references were rearranged.

  1. Response to comment: (Line 50: Arrange the references in ascending order of year here and also elsewhere in this draft such as Line 55, 160, 164.)

We are very sorry for our carelessness. These references were rearranged.

  1. Response to comment: (Line 72: Table 1; Arrange the events in ascending order of year and which magnitude are you taking, please provide full name, not just magnitude?)

The authors are very thankful for the viewer’s suggestion. The events were rearranged in ascending order of year. The magnitude mentioned in current text are moment magnitude, as can be noted the unit of the magnitude is .

Considering the viewer’s suggestion, the full name of the magnitude was provided as moment magnitude

  1. Response to comment: (Line 77: “The magnitude of …... rather than a constant value.” Rewrite this sentence.)

We are sorry for our unqualified English level. These sentences have been rewritten.

  1. Response to comment: (Line 85-96: This is irrelevant content, authors can remove it.)

We are very sorry for our carelessness. And this carelessness was corrected.

  1. Response to comment: (Line 112 and 114 and 149: “where ??? is the distribution…. And seismic zone *** and where v xxx is the average. Provide the missing content (???, ***, xxx).)

We are very sorry for this inconvenient to the reviewer. This is because the variables were mis-displayed. The possible reason is the usage of the ‘symbol’ font in the MS word. The corresponding text was corrected.

The variables and equations throughout the entire manuscript have been rewritten.

  1. Response to comment: (Line 131: It is related to the location …. conditions of the fault. Rewrite this sentence.)

We are sorry for our unqualified English level. These sentences have been rewritten.

  1. Response to comment: (Line 152: the heading of section 3 is the same as section 2. Please check it.)

We are very sorry for our carelessness. And this carelessness was corrected.

  1. Response to comment: (Line 154-156: “Located in the eastern margin …. types and activities. Rewrite this sentence)

We are sorry for our unqualified English level. These sentences have been rewritten.

  1. Response to comment: (Line 163: “are several … several meters”. Rewrite this sentence.)

We are very sorry for our carelessness, and the repeating sentence in the last paragraph of Introduction in were removed.

  1. Response to comment: (Line 168: In Figure 1, the Beijing city symbol not visible and provides the direction symbol as it is showing the map.)

We are very sorry for our carelessness. The symbol for Beijing was highlighted, and the direction symbol was provided in Fig. 1.

  1. Response to comment: (Line 172: Table 2; Arrange the events in ascending order of year and which magnitude are you taking, please provide full name, not just magnitude? Provide the references after Table 2 heading in ascending order of the year.)

The magnitude adopted throughout the manuscript is always moment magnitude, as can be noted the unit of the magnitude is . Considering the viewer’s suggestion, the full name of the magnitude was provided as moment magnitude in Tab. 2.

The events were rearranged in ascending order of year.

  1. Response to comment: (Line 174: heading is not relevant to main content; should be modified)

The authors are very thankful for the viewer’s suggestion. The heading of Sec. 3.2 was modified as “Seismic parameters of the example site”

  1. Response to comment: (Line 193: “Thus, the @@ is 1.74... missing content. Please provide)

We are very sorry for this inconvenient to the reviewer. This is because the variables were mis-displayed. The possible reason is the usage of the ‘symbol’ font in the MS word. The corresponding text was corrected.

The variables and equations throughout the entire manuscript have been rewritten.

  1. Response to comment: (Line 201: According to Ma et al. (2015) ….)

We are very sorry for our carelessness, and related text was corrected.

  1. Response to comment: (Line 205: In figure 3, authors can use the polynomial curve fit, and the equation of y can be changed. The earthquake magnitude is which magnitude, moment, or something else please specify.)

The authors have to say the usage of linear fitting here is to simplify the formula derivation work in the following text. When using polynomial fitting, the difficulty of formula derivation would be largely increased.

The author admits that this is one of the limitations of current work.

And the full name of the magnitude was provided as moment magnitude in Fig. 3.

  1. Response to comment: (Line 263: remove “4” in the last)

We are very sorry for our carelessness, and the related text was corrected.

  1. Response to comment: (Line 279: In figure 5, authors can use different line patterns (dotted, dash-dotted) for the graph to make it clearer in black and white, and what is P? Please write in the graph title.)

All due respect, the authors consider the current color mode is better, since in other figures such as Fig. 6 and 7, 6 groups of data were involved. It would difficult to distinguish them in grayscale mode.

And P stands for exceedance probability level, the related figure titles were modified to term this P.

  1. Response to comment: (Line 287: The reviewer is not getting the author's point. Please provide some background information.)

We are very sorry for our carelessness and our unqualified English level. The related text was improved.

  1. Response to comment: (Line 295: Remove the reference and provide after Table 6 heading in ascending order of the year.)

The authors are very thankful for the viewer’s suggestion. The related text was modified.

  1. Response to comment: (Line 299: Revised Figure 6 as suggested for Figure 5.)

All due respect, the authors consider the current color mode is better, since in Fig. 6 and 7, 6 groups of data were involved. It would difficult to distinguish them in grayscale mode.

  1. Response to comment: (Line 302: b should be in symbol)

We are very sorry for our carelessness; this mistake was corrected.

  1. Response to comment: (Line 319: remove the line “Probability level of 1-year and 50-year are taken as the study objects” as already stated at Line 300.)

This sentence was deleted.

  1. Response to comment: (Line 356: Point 6 is not a conclusion. It can be a summary of all conclusions so remove point 6.)

The authors are very thankful for the viewer’s suggestion. The point 6 was deleted.

Finally, the authors would special thanks to the viewer #1 for his/her good comments.

Reviewer 2 Report

In this work, the authors consider the construction of tunnels that more constantly are being built across active faults. The authors emphasize the importance that the magnitude of the fault dislocation has in the construction of these tunnels, so their proposal consists of presenting an improvement in the estimation of this parameter because, as they explain, the common practice is to consider this parameter in a deterministic way. Such improvement consists of using a probabilistic analysis in such a way as to consider the influence of seismic events (which can be reported in terms of seismic risk, that is, through the use of probabilities). They also apply their method to the case of the Xianglushan tunnel with the effect of the Longpan-qiaohou active fault.

The article is well presented and it seems to me that its contribution is clear as well of general interest since the proposed method can of course be applied in this type of construction (tunnels near active plates) in the rest of the world. I consider that this article can represent a valuable contribution therefore I recommend its publication after minor corrections, which I indicate below:

line 112: Missing word (variable): "where [ missing word ] is the distribution function [...]"
line 114: Missing word (variable) before "can be derived [...]"
line 143: Missing word (variable) before "is the magnitude distribution [...]"
line 149: Remove space before "is the average [...]", also there is also a missing word (the variable U) before "It is a number of [...]"
line 192: Missing word (variable) between "Thus, the [missing] is [...]"
line 113: Are the labels m_1 and m_0 of Table 3 in the right order? (because if so, it would mean that in line 198, M belongs to the range at 8.0<=m<=6.5?
Figure 3: Please explain somewhere why do you use a linear fit (y=47.5m-300) instead of using another like log-normal fit.
line 240: There is a "5" at the end of the sentence that I suppose is a typo, if so please erase, or correct it.
line 302-305: Maybe for future work, it would be interesting a correlation study of this observed behavior.
line 328-329: Please after you say "[...] the magnitude of fault dislocation is the important input parameter." write a short explanation that supports this statement.

-Erase indentation in lines 112, 114, 126, 137, and 149 (please check if there is another place where it applies).
-Some equations appear in a higher position with respect to the rest of the line, for instance in lines 137 and 142, please correct this.

Author Response

First of all, the authors are grateful for the reviewers’ comments on the language problem of the current manuscript. The authors are very sorry for this. And the author would have the manuscript been language edited by MDPI in the subsequent process, after the peer review stage.

 Response to Reviewer 2 Comments

  1. Response to comment: (line 112: Missing word (variable): "where [ missing word ] is the distribution function [...]")

We are very sorry for this inconvenient to the reviewer. This is because the variables were mis-displayed. The possible reason is the usage of the ‘symbol’ font in the MS word. The corresponding text was corrected.

The variables and equations throughout the entire manuscript have been rewritten.

  1. Response to comment: (line 114: Missing word (variable) before "can be derived [...]")

The variables and equations throughout the entire manuscript have been rewritten.

  1. Response to comment: (line 143: Missing word (variable) before "is the magnitude distribution [...]")

The variables and equations throughout the entire manuscript have been rewritten.

  1. Response to comment: (line 149: Remove space before "is the average [...]", also there is also a missing word (the variable U) before "It is a number of [...]")

The variables and equations throughout the entire manuscript have been rewritten.

  1. Response to comment: (line 192: Missing word (variable) between "Thus, the [missing] is [...]")

The variables and equations throughout the entire manuscript have been rewritten.

  1. Response to comment: (line 113: Are the labels m_1 and m_0 of Table 3 in the right order? (because if so, it would mean that in line 198, M belongs to the range at 8.0<=m<=6.5?)

We are very sorry for our carelessness. And this carelessness was corrected.

  1. Response to comment: (Figure 3: Please explain somewhere why do you use a linear fit (y=47.5m-300) instead of using another like log-normal fit.)

The authors have to say the usage of linear fitting here is to simplify the formula derivation work in the following text. When using polynomial fitting, the difficulty of formula derivation would be largely increased.

The author admits that this is one of the limitations of current work.

  1. Response to comment: (line 240: There is a "5" at the end of the sentence that I suppose is a typo, if so please erase, or correct it.)

We are very sorry for our carelessness. And this carelessness was corrected.

  1. Response to comment: (line 302-305: Maybe for future work, it would be interesting a correlation study of this observed behavior)

The authors are very thankful for the viewer’s suggestion. These works would be included in the authors’ future work.

  1. Response to comment: (line 328-329: Please after you say "[...] the magnitude of fault dislocation is the important input parameter." write a short explanation that supports this statement.)

The authors are very thankful for the viewer’s suggestion. Short explanation that why the magnitude of fault dislocation is the important for the anti-dislocation design of a tunnel is added.

  1. Response to comment: (-Erase indentation in lines 112, 114, 126, 137, and 149 (please check if there is another place where it applies).)

We are very sorry for our carelessness; these portions were modified.

  1. Response to comment: (-Some equations appear in a higher position with respect to the rest of the line, for instance in lines 137 and 142, please correct this.)

The variables and equations throughout the entire manuscript have been rewritten.

Finally, the authors would special thanks to the viewer #2 for his/her good comments.

Reviewer 3 Report

The article entitled "Probabilistic analysis of seismic hazard for the magnitude of fault dislocation induced by strong earthquakes: a case study from the Sichuan-Yunnan region" deals with an interesting topic, however it is not properly written and has a very sloppy presentation . It is also necessary to improve the English style and clearly define what is new in the article.

Some comments can help improve the article:

- The writing style is very repetitive. The ideas are constantly repeated changing the expression but saying the same thing. An example of this is the summary: Read carefully and you will see that the same ideas are repeated in short sentences and also repeat words constantly (eg in the first three lines the word “region” is constantly repeated). The writing style must be thoroughly reviewed.

- The presentation of the equations is chaotic. Formulas are not aligned and appear skipping lines.

- The symbology is also chaotic. Equation 3, for example, introduces the text "a site affected by dislocation magnitude of ..." and "a site affected by fault dislocation ..." in the equation itself, which makes the formula extend to two lines and is difficult to interpret. A nomenclature must be entered to reference these phrases. Furthermore, the structure of the formula presented is indecipherable. Where is the integrand? Why is the product sometimes symbolized with "x" and sometimes with nothing? What does "dedudm" mean? Before submitting the article, take a few minutes to present it properly.

- Authors read lines 85-96. How can the article be presented without previously reviewing the text? We can all be wrong but the lack of care in the editing of the article is evident.

- There are symbols that disappear from the text, for example in line 112, 114, 192,….

- There are symbols whose meaning is not explained in the text, such as Eu, S, Smax, N,….

- Equations (3) to (6) are all from Su et al. (1993). This must be clearly stated. Therefore, it is also clear that the formulation is not new, what is the novelty of the article? I think clearly the article focuses on the application to the Sichuan-Yunnan tunnel. That is, no new formulation or methodology is provided and in the end the application is made, but the essence of the article is the application itself. This should be reflected in the abstract, the introduction, objectives and conclusions of the article, so that the paper is rewritten, focusing on what is truly contributed and presenting the formulation as state of the art and methodology used but without novelty purposes.

- Lines 192 to 195 do not adequately explain obtaining parameters. The explanation should be rewritten and expanded.

- Fig. 3 inserted in the text is not well understood, What do the two lines represented mean?

- Section 3.4 is presented as an unstructured script; it should be better written and explained taking care once more of the presentation format.

- Parameters a, b, c and d in section 3.4 are not adequately explained as they are derived. It must be detailed.

- Line 240 "Here, the random error and is neglected in the calculation.5" what is that? Why is it neglected?

- Line 263 “Republic of China, 2018) .4” What's that?

- Figure 4 is the same as Figure 3, why are they repeated? What space do you have in mind for each one in the text?

- Section 4.1 to 4.3 presents a sensitivity analysis of certain parameters, but its results are not discussed in relation to application. It should be discussed what is the proper range of our analysis. In this sensitivity study, the great variation of probability results is observed and as for certain values of the parameters there are abrupt changes in the probability trends. This implies a great sensitivity of parameters and therefore a reduction in the reliability of the calculations. How do you introduce this reliability in the analysis of the application? These aspects are key and must be discussed.

- The conclusions must be rewritten with the true contribution of the article and eliminated what is already known. For example, point (1) is already known, the authors have not discovered it. Point (2) should also not be indicated as a contribution because the formulation is already known.

Author Response

First of all, the authors are grateful for the reviewers’ comments on the language problem of the current manuscript. The authors are very sorry for this. And the author would have the manuscript been language edited by MDPI in the subsequent process, after the peer review stage.

 Response to Reviewer 3 Comments

  1. Response to comment: (…The writing style must be thoroughly reviewed.)

We are sorry for our unqualified English level. The authors did their best to improve the writing style. And the author would have the manuscript been language edited in the subsequent process, after the peer review stage.

  1. Response to comment: (- The presentation of the equations is chaotic. Formulas are not aligned and appear skipping lines……)

The variables and equations throughout the entire manuscript have been rewritten.

  1. Response to comment: (The symbology is also chaotic……)

We are very sorry for this inconvenient to the reviewer. This is because the variables were mis-displayed. The possible reason is the usage of the ‘symbol’ font in the MS word. The corresponding text was corrected.

The variables and equations throughout the entire manuscript have been rewritten.

  1. Response to comment: (- Authors read lines 85-96. How can the article be presented without previously reviewing the text? We can all be wrong but the lack of care in the editing of the article is evident.)

We are very sorry for our carelessness, and related text was corrected.

  1. Response to comment: (- There are symbols that disappear from the text, for example in line 112, 114, 192,….)

We are very sorry for this inconvenient to the reviewer. This is because the variables were mis-displayed. The possible reason is the usage of the ‘symbol’ font in the MS word. The corresponding text was corrected.

The variables and equations throughout the entire manuscript have been rewritten.

  1. Response to comment: (There are symbols whose meaning is not explained in the text, such as Eu, S, Smax, N,….)

We are very sorry for our carelessness, descriptive text was added for these variables.

  1. Response to comment: (- Equations (3) to (6) are all from Su et al. (1993). This must be clearly stated…)

The authors are very thankful for the viewer’s suggestion. Reference of Su et al. (1993) was clearly stated in the current manuscript.

And yes! the main focus of the manuscript was about the application of the PSHA-based dislocation estimation to the Sichuan-Yunnan tunnel. However, some special local seismic conditions and parameters were considered in the application process, as can be found in Tab.5.

Considering the reviewer’s suggestion, the abstract and conclusion were revised.

  1. Response to comment: (- Lines 192 to 195 do not adequately explain obtaining parameters. The explanation should be rewritten and expanded.)

The related text has been rewritten

  1. Response to comment: (- Fig. 3 inserted in the text is not well understood, What do the two lines represented mean?)

Sorry for this confusion to the reviewer, the solid line is the probability function of fault dislocation under various seismic magnitude for the western region of China by Ma et al. (2005). While the dashed line is the simplified version adopted in this study.

The related explanation test was improved.

  1. Response to comment: (- Section 3.4 is presented as an unstructured script; it should be better written and explained taking care once more of the presentation format.)

The Sec. 3.4 has been rewritten.

  1. Response to comment: (- Parameters a, b, c and d in section 3.4 are not adequately explained as they are derived. It must be detailed.)

The Sec. 3.4 has been rewritten.

  1. Response to comment: (- Line 240 "Here, the random error and is neglected in the calculation.5" what is that? Why is it neglected?)

We are sorry for our unqualified English level. It should be ‘ignored’, rather than ‘neglected’. The random error e is ignored here for concision and simplicity.

  1. Response to comment: (Line 263 “Republic of China, 2018) .4” What's that?)

Sorry for this typo, it was corrected.

  1. Response to comment: (- Figure 4 is the same as Figure 3, why are they repeated? What space do you have in mind for each one in the text?)

We are very sorry for our carelessness; Fig. 4 was corrected.

  1. Response to comment: (- Section 4.1 to 4.3 presents a sensitivity analysis of certain parameters,…)

The authors are very thankful for the viewer’s suggestion. Section 4.1 to 4.3 were improved, whit more discussions were introduced in Sec. 4.1 to 4.3.

  1. Response to comment: (- The conclusions must be rewritten with the true contribution…)

The abstract and conclusion were revised.

Finally, the authors would special thanks to the viewer #3 for his/her good comments.

Round 2

Reviewer 1 Report

I am pleased to read the draft titled “Probabilistic seismic hazard analysis for fault dislocation magnitude induced by strong earthquakes: a case study of the Sichuan-Yunnan region” and share my opinions.

The reviewer is fully convinced with the current draft and technical novelty beyond prior published papers in this area. The reviewer is happy to recommend this draft for publication. However, please check the whole draft again for an English check.

Reviewer 3 Report

No comments. Accept

This manuscript is a resubmission of an earlier submission. The following is a list of the peer review reports and author responses from that submission.